# Adaptive stimulus selection for optimizing neural population responses

**Benjamin R. Cowley**[1,2], **Ryan C. Williamson**[1,2,5], **Katerina Acar**[2,6],
**Matthew A. Smith**[*,2,7], **Byron M. Yu**[*,2,3,4]
[1]Machine Learning Dept., [2]Center for Neural Basis of Cognition, [3]Dept. of Electrical
and Computer Engineering, [4]Dept. of Biomedical Engineering, Carnegie Mellon University
[5]School of Medicine, [6]Dept. of Neuroscience, [7]Dept. of Ophthalmology, University of Pittsburgh
bcowley@cs.cmu.edu, {rcw30, kac216, smithma}@pitt.edu, byronyu@cmu.edu
[*]denotes equal contribution.

## Abstract

Adaptive stimulus selection methods in neuroscience have primarily focused on maximizing the firing rate of a single recorded neuron. When recording from a population of neurons, it is usually not possible to find a single stimulus that maximizes the firing rates of all neurons. This motivates optimizing an objective function that takes into account the responses of all recorded neurons together. We propose "Adept," an adaptive stimulus selection method that can optimize population objective functions. In simulations, we first confirmed that population objective functions elicited more diverse stimulus responses than single-neuron objective functions. Then, we tested Adept in a closed-loop electrophysiological experiment in which population activity was recorded from macaque V4, a cortical area known for mid-level visual processing. To predict neural responses, we used the outputs of a deep convolutional neural network model as feature embeddings. Natural images chosen by Adept elicited mean neural responses that were 20% larger than those for randomly-chosen natural images, and also evoked a larger diversity of neural responses. Such adaptive stimulus selection methods can facilitate experiments that involve neurons far from the sensory periphery, for which it is often unclear which stimuli to present.

## 1 Introduction

A key choice in a neurophysiological experiment is to determine which stimuli to present. Often, it is unknown *a priori* which stimuli will drive a to-be-recorded neuron, especially in brain areas far from the sensory periphery. Most studies either choose from a class of parameterized stimuli (e.g., sinusoidal gratings or pure tones) or present many randomized stimuli (e.g., white noise) to find the stimulus that maximizes the response of a neuron (i.e., the preferred stimulus) [1, 2]. However, the first approach limits the range of stimuli explored, and the second approach may not converge in a finite amount of recording time [3]. To efficiently find a preferred stimulus, studies have employed adaptive stimulus selection (also known as "adaptive sampling" or "optimal experimental design") to determine the next stimulus to show given the responses to previous stimuli in a closed-loop experiment [4, 5]. Many adaptive methods have been developed to find the smallest number of stimuli needed to fit parameters of a model that predicts the recorded neuron's activity from the stimulus [6, 7, 8, 9, 10, 11]. When no encoding model exists for a neuron (e.g., neurons in higher visual cortical areas), adaptive methods rely on maximizing the neuron's firing rate via genetic algorithms [12, 13, 14] or gradient ascent [15, 16] to home in on the neuron's preferred stimulus. To our knowledge, all current adaptive stimulus selection methods focus solely on optimizing the firing rate of a *single* neuron.

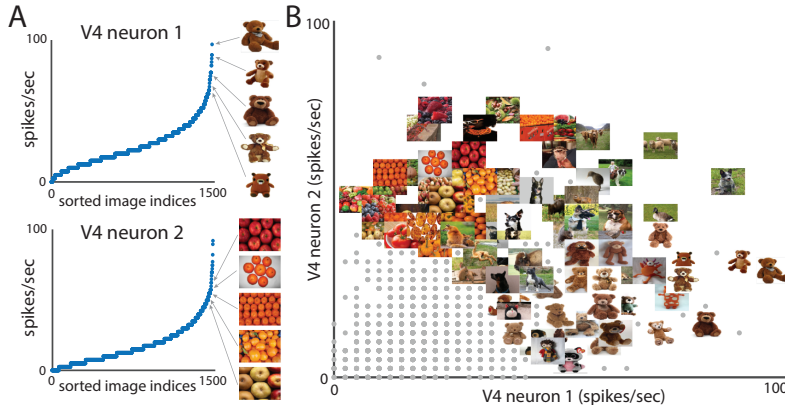

Figure 1: Responses of two macaque V4 neurons. *A*. Different neurons prefer different stimuli. Displayed images evoked 5 of top 25 largest responses. *B*. Images placed according to their responses. Gray dots represent responses to other images. Same neurons as in *A*.

Developments in neural recording technologies now enable the simultaneous recordings of tens to hundreds of neurons [17], each of which has its own preferred stimulus. For example, consider two neurons recorded in V4, a mid-level visual cortical area (Fig. 1*A*). Whereas neuron 1 responds most strongly to teddy bears, neuron 2 responds most strongly to arranged circular fruit. Both neurons moderately respond to images of animals (Fig. 1*B*). Given that different neurons have different preferred stimuli, how do we select which stimuli to present when simultaneously recording from multiple neurons? This necessitates defining objective functions for adaptive stimulus selection that are based on a population of neurons rather than any single neuron. Importantly, these objective functions can go beyond simply maximizing the firing rates of neurons and instead can be optimized for other attributes of the population response, such as maximizing the scatter of the responses in a multi-neuronal response space (Fig. 1*B*).

We propose Adept, an adaptive stimulus selection method that "adeptly" chooses the next stimulus to show based on a population objective function. Because the neural responses to candidate stimuli are unknown, Adept utilizes feature embeddings of the stimuli to predict to-be-recorded responses. In this work, we use the feature embeddings of a deep convolutional neural network (CNN) for prediction. We first confirmed with simulations that Adept, using a population objective function, elicited larger mean responses and a larger diversity of responses than optimizing the response of each neuron separately. Then, we ran Adept on V4 population activity recorded during a closed-loop electrophysiological experiment. Images chosen by Adept elicited higher mean firing rates and more diverse population responses compared to randomly-chosen images. This demonstrates that Adept is effective at finding stimuli to drive a population of neurons in brain areas far from the sensory periphery.

## 2   Population objective functions

Depending on the desired outcomes of an experiment, one may favor one objective function over another. Here we discuss different objection functions for adaptive stimulus selection and the resulting responses $\mathbf{r} \in \mathbb{R}^p$, where the $i$th element $\mathbf{r}^i$ is the response of the $i$th neuron ($i = 1, \ldots, p$) and $p$ is the number of neurons recorded simultaneously. To illustrate the effects of different objective functions, we ran an adaptive stimulus selection method on the activity of two simulated neurons (see details in Section 5.1). We first consider a single-neuron objective function employed by many adaptive methods [12, 13, 14]. Using this objective function $f(\mathbf{r}) = \mathbf{r}^i$, which maximizes the response for the $i$th neuron of the population, the adaptive method for $i = 1$ chose stimuli that maximized neuron 1's response (Fig. 2*A*, red dots). However, images that produced large responses for neuron 2 were not chosen (Fig. 2*A*, top left gray dots).

A natural population-level extension to this objective function is to maximize the responses of all neurons by defining the objective function to be $f(\mathbf{r}) = \|\mathbf{r}\|_2$. This objective function led to choosing stimuli that maximized responses for neurons 1 and 2 individually, as well as large responses for both neurons together (Fig. 2*B*). Another possible objective function is to maximize the scatter of the responses. In particular, we would like to choose the next stimulus such that the response vector $\mathbf{r}$ is far away from the previously-seen response vectors $\mathbf{r}_1, \ldots, \mathbf{r}_M$ after $M$ chosen stimuli. One way to achieve this is to maximize the average Euclidean distance between $\mathbf{r}$ and $\mathbf{r}_1, \ldots, \mathbf{r}_M$, which leads

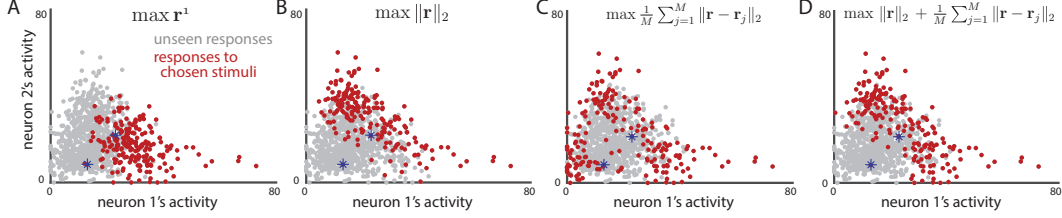

Figure 2: Different objective functions for adaptive stimulus selection yield different observed population responses (red dots). Blue * denote responses to stimuli used to initialize the adaptive method (the same for each panel).

to the objective function $f(\mathbf{r}, \mathbf{r}_1, \ldots, \mathbf{r}_M) = \frac{1}{M} \sum_{j=1}^{M} \|\mathbf{r} - \mathbf{r}_j\|_2$. This objective function led to a large scatter in responses for neurons 1 and 2 (Fig. 2C, red dots near and far from origin). This is because choosing stimuli that yield small and large responses produces the largest distances between responses.

Finally, we considered an objective function that favored large responses that are far away from one another. To achieve this, we summed the objectives in Fig. 2B and 2C. The objective function $f(\mathbf{r}, \mathbf{r}_1, \ldots, \mathbf{r}_M) = \|\mathbf{r}\|_2 + \frac{1}{M} \sum_{j=1}^{M} \|\mathbf{r} - \mathbf{r}_j\|_2$ was able to uncover large responses for both neurons (Fig. 2D, red dots far from origin). It also led to a larger scatter than maximizing the norm of $\mathbf{r}$ alone (e.g., compare red dots in bottom right of Fig. 2B and Fig. 2D). For these reasons, we use this objection function in the remainder of this work. However, the Adept framework is general and can be used with many different objective functions, including all presented in this section.

## 3  Using feature embeddings to predict norms and distances

We now formulate the optimization problem using the last objective function in Section 2. Consider a pool of $N$ candidate stimuli $\mathbf{s}_1, \ldots, \mathbf{s}_N$. After showing $(t - 1)$ stimuli, we are given previously-recorded response vectors $\mathbf{r}_{n_1}, \ldots, \mathbf{r}_{n_{t-1}} \in \mathbb{R}^p$, where $n_1, \ldots, n_{t-1} \in \{1, \ldots, N\}$. In other words, $\mathbf{r}_{n_j}$ is the vector of responses to the stimulus $\mathbf{s}_{n_j}$. At the $t$th iteration of adaptive stimulus selection, we choose the index $n_t$ of the next stimulus to show by the following:

$$n_t = \underset{s \in \{1, \ldots, N\} \setminus \{n_1, \ldots, n_{t-1}\}}{\arg\max} \|\mathbf{r}_s\|_2 + \frac{1}{t-1} \sum_{j=1}^{t-1} \|\mathbf{r}_s - \mathbf{r}_{n_j}\|_2 \qquad (1)$$

where $\mathbf{r}_s$ is the unseen population response vector to stimulus $\mathbf{s}_s$.

If the $\mathbf{r}_s$ were known, we could directly optimize Eqn. 1. However, in an online setting, we do not have access to the $\mathbf{r}_s$. Instead, we can directly predict the norm and average distance terms in Eqn. 1 by relating distances in neural response space to distances in a feature embedding space. The key idea is that if two stimuli have similar feature embeddings, then the corresponding neural responses will have similar norms and average distances. Concretely, consider feature embedding vectors $\mathbf{x}_1, \ldots, \mathbf{x}_N \in \mathbb{R}^q$ corresponding to candidate stimuli $\mathbf{s}_1, \ldots, \mathbf{s}_N$. For example, we can use the activity of $q$ neurons from a CNN as a feature embedding vector for natural images [18]. To predict the norm of unseen response vector $\mathbf{r}_s \in \mathbb{R}^p$, we use kernel regression with the previously-recorded response vectors $\mathbf{r}_{n_1}, \ldots, \mathbf{r}_{n_{t-1}}$ as training data [19]. To predict the distance between $\mathbf{r}_s$ and a previously-recorded response vector $\mathbf{r}_{n_j}$, we extend kernel regression to account for the paired nature of distances. Thus, the norm and average distance in Eqn. 1 for the unseen response vector $\mathbf{r}_s$ to the $s$th candidate stimulus are predicted by the following:

$$\widehat{\|\mathbf{r}_s\|_2} = \sum_k \frac{K(\mathbf{x}_s, \mathbf{x}_{n_k})}{\sum_\ell K(\mathbf{x}_s, \mathbf{x}_{n_\ell})} \|\mathbf{r}_{n_k}\|_2, \qquad \widehat{\|\mathbf{r}_s - \mathbf{r}_{n_j}\|_2} = \sum_k \frac{K(\mathbf{x}_s, \mathbf{x}_{n_k})}{\sum_\ell K(\mathbf{x}_s, \mathbf{x}_{n_\ell})} \|\mathbf{r}_{n_k} - \mathbf{r}_{n_j}\|_2$$

$$(2)$$

where $k, \ell \in \{1, \ldots, t-1\}$. Here we use the radial basis function kernel $K(\mathbf{x}_j, \mathbf{x}_k) = \exp(-\|\mathbf{x}_j - \mathbf{x}_k\|_2^2 / h^2)$ with kernel bandwidth $h$, although other kernels can be used.

We tested the performance of this approach versus three other possible prediction approaches. The first two approaches use linear ridge regression and kernel regression, respectively, to predict $\mathbf{r}_s$. Their

prediction $\hat{\mathbf{r}}_s$ is then used to evaluate the objective in place of $\mathbf{r}_s$. The third approach is a linear ridge regression version of Eqn. 2 to directly predict $\|\mathbf{r}_s\|_2$ and $\|\mathbf{r}_s - \mathbf{r}_{n_j}\|_2$. To compare the performance of these approaches, we developed a testbed in which we sampled two distinct populations of neurons from the same CNN, and asked how well one population can predict the responses of the other population using the different approaches described above. Formally, we let $\mathbf{x}_1, \ldots, \mathbf{x}_N$ be feature embedding vectors of $q = 500$ CNN neurons, and response vectors $\mathbf{r}_{n_1}, \ldots, \mathbf{r}_{n_{800}}$ be the responses of $p = 200$ different CNN neurons to 800 natural images. CNN neurons were from the same GoogLeNet CNN [18] (see CNN details in Results). To compute performance, we took the Pearson's correlation $\rho$ between the predicted and actual objective values on a held out set of responses not used for training. We also tracked the computation time $\tau$ (computed on an Intel Xeon 2.3GHz CPU with 36GB RAM) because these computations need to occur between stimulus presentations in an electrophysiological experiment. The approach in Eqn. 2 performed the best ($\rho = 0.64$) and was the fastest ($\tau = 0.2$ s) compared to the other prediction approaches ($\rho = 0.39, 0.41, 0.23$ and $\tau = 12.9$ s, 1.5 s, 48.4 s, for the three other approaches, respectively). The remarkably faster speed of Eqn. 2 over other approaches comes from the evaluation of the objective function (fast matrix operations), the fact that no training of linear regression weight vectors is needed, and the fact that distances are directly predicted (unlike the approaches that first predict $\hat{\mathbf{r}}_s$ and then must re-compute distances between $\hat{\mathbf{r}}_s$ and $\mathbf{r}_{n_1}, \ldots, \mathbf{r}_{n_{t-1}}$ for each candidate stimulus $s$). Due to its performance and fast computation time, we use the prediction approach in Eqn. 2 for the remainder of this work.

## 4 Adept algorithm

We now combine the optimization problem in Eqn. 1 and prediction approach in Eqn. 2 to formulate the Adept algorithm. We first discuss the adaptive stimulus selection paradigm (Fig. 3, left) and then the Adept algorithm (Fig. 3, right).

For the adaptive stimulus selection paradigm (Fig. 3, left), the experimenter first selects a candidate stimulus pool $\mathbf{s}_1, \ldots, \mathbf{s}_N$ from which Adept chooses, where $N$ is large. For a vision experiment, the candidate stimulus pool could comprise natural images, textures, or sinusoidal gratings. For an auditory experiment, the stimulus pool could comprise natural sounds or pure tones. Next, feature embedding vectors $\mathbf{x}_1, \ldots, \mathbf{x}_N \in \mathbb{R}^q$ are computed for each candidate stimulus, and the pre-computed $N \times N$ kernel matrix $K(\mathbf{x}_j, \mathbf{x}_k)$ (i.e., similarity matrix) is input into Adept. For visual neurons, the feature embeddings could come from a bank of Gabor-like filters with different orientations and spatial frequencies [20], or from a more expressive model, such as CNN neurons in a middle layer of a pre-trained CNN. Because Adept only takes as input the kernel matrix $K(\mathbf{x}_j, \mathbf{x}_k)$ and not the feature embeddings $\mathbf{x}_1, \ldots, \mathbf{x}_N$, one could alternatively use a similarity matrix computed from psychophysical data to define the similarity between stimuli if no model exists. The previously-recorded response vectors $\mathbf{r}_{n_1}, \ldots, \mathbf{r}_{n_{t-1}}$ are also input into Adept, which then outputs the next chosen stimulus $\mathbf{s}_{n_t}$ to show. While the observer views $\mathbf{s}_{n_t}$, the response vector $\mathbf{r}_{n_t}$ is recorded and appended to the previously-recorded response vectors. This procedure is iteratively repeated until the end of the recording session. To show as many stimuli as possible, Adept does not choose the same stimulus more than once.

For the Adept algorithm (Fig. 3, right), we initialize by randomly choosing a small number of stimuli (e.g., $N_{\text{init}} = 5$) from the large pool of $N$ candidate stimuli and presenting them to the observer. Using the responses to these stimuli $\mathbf{R}(:, 1{:}N_{\text{init}})$, Adept then adaptively chooses a new stimulus by finding the candidate stimulus that yields the largest objective (in this case, using the objective defined by Eqns. 1 and 2). This search is carried out by evaluating the objective for every candidate stimulus. There are three primary reasons why Adept is computationally fast enough to consider all candidate stimuli. First, the kernel matrix $K_{\mathbf{X}}$ is pre-computed, which is then easily indexed. Second, the prediction of the norm and average distance is computed with fast matrix operations. Third, Adept updates the distance matrix $D_{\mathbf{R}}$, which contains the pairwise distances between recorded response vectors, instead of re-computing $D_{\mathbf{R}}$ at each iteration.

## 5 Results

We tested Adept in two settings. First, we tested Adept on a surrogate for the brain—a pre-trained CNN. This allowed us to perform comparisons between methods with a noiseless system. Second, in a closed-loop electrophysiological experiment, we performed Adept on population activity recorded in macaque V4. In both settings, we used the same candidate image pool of $N \approx 10{,}000$ natural

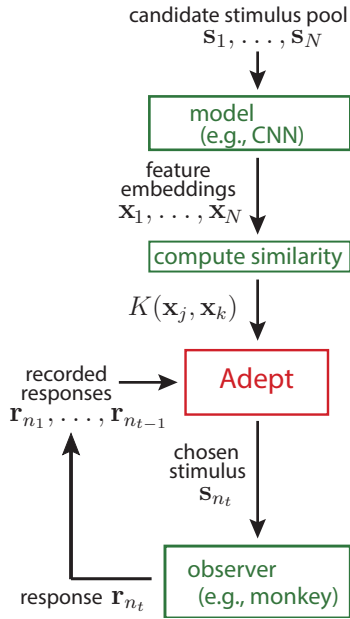

**Algorithm 1:** Adept algorithm

---

**Input:** $N$ candidate stimuli, feature embeddings $\mathbf{X}(q \times N)$, kernel bandwidth $h$ (hyperparameter)

**Initialization:**
$\quad K_{\mathbf{X}}(j,k) = \exp(-\|\mathbf{X}(:,j) - \mathbf{X}(:,k)\|_2^2/h^2)$ for all $j, k$
$\quad \mathbf{R}(:, 1{:}N_{\text{init}}) \leftarrow$ responses to $N_{\text{init}}$ initial stimuli
$\quad D_{\mathbf{R}}(j,k) = \|\mathbf{R}(:,j) - \mathbf{R}(:,k)\|_2$ for $j, k = 1, \ldots, N_{\text{init}}$
$\quad \text{ind\_obs} \leftarrow$ indices of $N_{\text{init}}$ observed stimuli

**Online algorithm:**
**for** $t$th stimulus to show **do**
$\quad$ **for** $s$th candidate stimulus **do**
$\quad\quad k_{\mathbf{X}} = K_{\mathbf{X}}(\text{ind\_obs}, s)/\sum_{\ell \in \text{ind\_obs}} K_{\mathbf{X}}(\ell, s)$
$\quad\quad$ % predict norm from recorded responses
$\quad\quad \text{norms}(s) \leftarrow \widehat{\|\mathbf{r}_s\|_2} = k_{\mathbf{X}}{}^T \text{diag}(\sqrt{\mathbf{R}^T\mathbf{R}})$
$\quad\quad$ % predict average distance from recorded responses
$\quad\quad \text{avgdists}(s) \leftarrow \frac{1}{t-1}\sum_\ell \widehat{\|\mathbf{r}_s - \mathbf{r}_{n_\ell}\|_2} = \text{mean}(k_{\mathbf{X}}{}^T D_{\mathbf{R}})$
$\quad$ **end**
$\quad \text{ind\_obs}(N_{\text{init}} + t) \leftarrow \text{argmax}(\text{norms} + \text{avgdists})$
$\quad \mathbf{R}(:, N_{\text{init}} + t) \leftarrow$ recorded responses to chosen stimulus
$\quad$ update $D_{\mathbf{R}}$ with $\|\mathbf{R}(:, N_{\text{init}} + t) - \mathbf{R}(:, \ell)\|_2$ for all $\ell$
**end**

Figure 3: Flowchart of the adaptive sampling paradigm (left) and the Adept algorithm (right).

images from the McGill natural image dataset [21] and Google image search [22]. For the predictive feature embeddings in both settings, we used responses from a pre-trained CNN different from the CNN used as a surrogate for the brain in the first setting. The motivation to use CNNs was inspired by the recent successes of CNNs to predict neural activity in V4 [23].

## 5.1 Testing Adept on CNN neurons

The testbed for Adept involved two different CNNs. One CNN is the surrogate for the brain. For this CNN, we took responses of $p = 200$ neurons in a middle layer of the pre-trained ResNet CNN [24] (layer 25 of 50, named 'res3dx'). A second CNN is used for feature embeddings to predict responses of the first CNN. For this CNN, we took responses of $q = 750$ neurons in a middle layer of the pre-trained GoogLeNet CNN [18] (layer 5 of 10, named 'icp4_out'). Both CNNs were trained for image classification but had substantially different architectures. Pre-trained CNNs were downloaded from MatConvNet [25], with the PVT version of GoogLeNet [26]. We ran Adept for 2,000 out of the 10,000 candidate images (with $N_{\text{init}} = 5$ and kernel bandwidth $h = 200$—similar results were obtained for different $h$), and compared the CNN responses to those of 2,000 randomly-chosen images. We asked two questions pertaining to the two terms in the objective function in Eqn. 1. First, are responses larger for Adept than for randomly-chosen images? Second, to what extent does Adept produce larger scatter of responses than if we had chosen images at random? A larger scatter implies a greater diversity in evoked population responses (Fig. 1*B*).

To address the first question, we computed the mean response across all 2,000 images for each CNN neuron. The mean responses using Adept were on average 15.5% larger than the mean responses to randomly chosen images (Fig. 4*A*, difference in means was significantly greater than zero, $p < 10^{-4}$). For the second question, we assessed the amount of response scatter by computing the amount of variance captured by each dimension. We applied PCA separately to the responses to images chosen by Adept and those to images selected randomly. For each dimension, we computed the ratio between the Adept eigenvalue divided by the randomly-chosen-image eigenvalue. In this way, we compared the dimensions of greatest variance, followed by the dimensions of the second-most variance, and so on. Ratios above 1 indicate that Adept explored a dimension more than the corresponding ordered dimension of random selection. We found that Adept produced larger response scatter compared to randomly-chosen images for many dimensions (Fig. 4*B*). Ratios for dimensions of lesser variance (e.g., dimensions 10 to 75) are nearly as meaningful as those of the dimensions of greatest variance

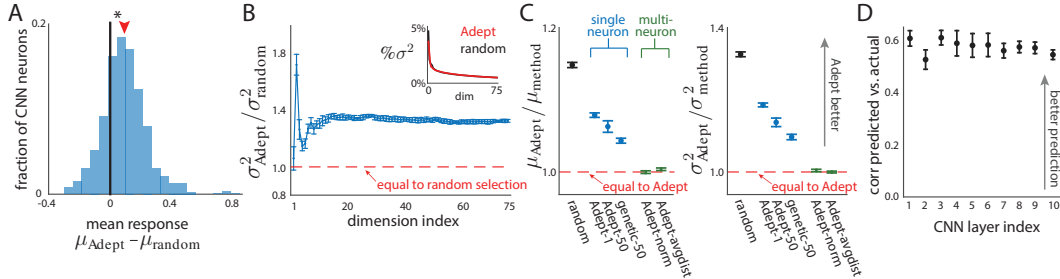

Figure 4: CNN testbed for Adept. *A*. Mean responses (arbitrary units) to images chosen by Adept were greater than to randomly-chosen images. *B*. Adept produced higher response variance for each PC dimension than when randomly choosing images. Inset: Percent variance explained. *C*. Relative to the full objective function in Eqn. 1, population objective functions (green) yielded higher response mean and variance than those of single-neuron objective functions (blue). *D*. Feature embeddings for all CNN layers were predictive. Error bars are $\pm$ s.d. across 10 runs.

(i.e., dimensions 1 to 10), as the top 10 dimensions explained only 16.8% of the total variance (Fig. 4*B*, inset).

Next, we asked to what extent does optimizing a population objective function perform better than optimizing a single-neuron objective function. For the single-neuron case, we implemented three different methods. First, we ran Adept to optimize the response of a single CNN neuron with the largest mean response ('Adept-1'). Second, we applied Adept in a sequential manner to optimize the response of 50 randomly-chosen CNN neurons individually. After optimizing a CNN neuron for 40 images, optimization switched to the next CNN neuron ('Adept-50'). Third, we sequentially optimized 50 randomly-chosen CNN neurons individually using a genetic algorithm ('genetic-50'), similar to the ones proposed in previous studies [12, 13, 14]. We found that Adept produced higher mean responses than the three single-neuron methods (Fig. 4*C*, blue points in left panel), likely because Adept chose images that evoked large responses across neurons together. All methods produced higher mean responses than randomly choosing images (Fig. 4*C*, black point above blue points in left panel). Adept also produced higher mean eigenvalue ratios across the top 75 PCA dimensions than the three single-neuron methods (Fig. 4*C*, blue points in right panel). This indicates that Adept, using a population objective, is better able to optimize population responses than using a single-neuron objective to optimize the response of each neuron in the population.

We then modified the Adept objective function to include only the norm term ('Adept-norm', Fig. 2*B*) and only the average distance term ('Adept-avgdist', Fig. 2*C*). Both of these population methods performed better than single-neuron methods (Fig. 4*C*, green points below blue points). While their performance was comparable to Adept using the full objective function, upon closer inspection, we observed differences in performance that matched our intuition about the objective functions. The mean response ratio for Adept using the full objection function and Adept-norm was close to 1 (Fig. 4*C*, left panel, Adept-norm on red-dashed line, $p = 0.65$), but the eigenvalue ratio was greater than 1 (Fig. 4*C*, right panel, Adept-norm above red-dashed line, $p < 0.005$). Thus, Adept-norm maximizes mean responses at the expense of less scatter. On the other hand, Adept-avgdist produced a lower mean response than that of Adept using the full objective function (Fig. 4*C*, left panel, Adept-avgdist above red-dashed line, $p < 10^{-4}$), but an eigenvalue ratio of 1 (Fig. 4*C*, right panel, Adept-avgdist on red-dashed line, $p = 0.62$). Thus, Adept-avgdist increases the response scatter at the expense of a lower mean response.

The results in this section were based on middle layer neurons in the GoogLeNet CNN predicting middle layer neurons in the ResNet CNN. However, it is possible that CNN neurons in other layers may be better predictors than those in a middle layer. To test for this, we asked which layers of the GoogLeNet CNN were most predictive of the objective values of the middle layer of the ResNet CNN. For each layer of increasing depth, we computed the correlation between the predicted objective (using 750 CNN neurons from that layer) and the actual objective of the ResNet responses (200 CNN neurons) (Fig. 4*D*). We found that all layers were predictive ($\rho \approx 0.6$), although there was variation across layers. Middle layers were slightly more predictive than deeper layers, likely because

deeper layers of GoogLeNet have a different embedding of natural images than the middle layer of the ResNet CNN.

## 5.2  Testing Adept on V4 population recordings

Next, we tested Adept in a closed-loop neurophysiological experiment. We implanted a 96-electrode array in macaque V4, whose neurons respond differently to a wide range of image features, including orientation, spatial frequency, color, shape, texture, and curvature, among others [27]. Currently, no existing parametric encoding model fully captures the stimulus-response relationship of V4 neurons. The current state-of-the-art model for predicting the activity of V4 neurons uses the output of middle layer neurons in a CNN previously trained without any information about the responses of V4 neurons [23]. Thus, we used a pre-trained CNN (GoogLeNet) to obtain the predictive feature embeddings.

The experimental task flow proceeded as follows. On each trial, a monkey fixated on a central dot while an image flashed four times in the aggregate receptive fields of the recorded V4 neurons. After the fourth flash, the monkey made a saccade to a target dot (whose location was unrelated to the shown image), for which he received a juice reward. During this task, we recorded threshold crossings on each electrode (referred to as "spikes"), where the threshold was defined as a multiple of the RMS voltage set independently for each channel. This yielded 87 to 96 neural units in each session. The spike counts for each neural unit were averaged across the four 100 ms flashes to obtain mean responses. The mean response vector for the $p$ neural units was then appended to the previously-recorded responses and input into Adept. Adept then output an image to show on the next trial. For the predictive feature embeddings, we used $q = 500$ CNN neurons in the fifth layer of GoogLeNet CNN (kernel bandwidth $h = 200$). In each recording session, the monkey typically performed 2,000 trials (i.e., 2,000 of the $N = 10,000$ natural images would be sampled). Each Adept run started with $N_{\text{init}} = 5$ randomly-chosen images.

We first recorded a session in which we used Adept during one block of trials and randomly chose images in another block of trials. To qualitatively compare Adept and randomly selecting images, we first applied PCA to the response vectors of both blocks, and plotted the top two PCs (Fig. 5A, left panel). Adept uncovers more responses that are far away from the origin (Fig. 5A, left panel, red dots farther from black * than black dots). For visual clarity, we also computed kernel density estimates for the Adept responses ($p_{\text{Adept}}$) and responses to randomly-chosen images ($p_{\text{random}}$), and plotted the difference $p_{\text{Adept}} - p_{\text{random}}$ (Fig. 5A, right panel). Responses for Adept were denser than for randomly-chosen images further from the origin, whereas the opposite was true closer to the origin (Fig. 5A, right panel, red region further from origin than black region). These plots suggest that Adept uncovers large responses that are far from one another. Quantitatively, we verified that Adept chose images with larger objective values in Eqn. 1 than randomly-chosen images (Fig. 5B). This result is not trivial because it relies on the ability of the CNN to predict V4 population responses. If the CNN predicted V4 responses poorly, the objective evaluated on the V4 responses to images chosen by Adept could be lower than that evaluated on random images.

We then compared Adept and random stimulus selection across 7 recording sessions, including the above session (450 trials per block, with three sessions with the Adept block before the random selection block, three sessions with the opposite ordering, and one session with interleaved trials). We found that the images chosen by Adept produced on average 19.5% higher mean responses than randomly-chosen images (Fig. 5C, difference in mean responses were significantly greater than zero, $p < 10^{-4}$). We also found that images chosen by Adept produced greater response scatter than for randomly-chosen images, as the mean ratios of eigenvalues were greater than 1 (Fig. 5D, dimensions 1 to 5). Yet, there were dimensions for which the mean ratios of eigenvalues were less than 1 (Fig. 5D, dimensions 9 and 10). These dimensions explained little overall variance ($< 5\%$ of the total response variance).

Finally, we asked to what extent do the different CNN layers predict the objective of V4 responses, as in Fig. 4D. We found that, using 500 CNN neurons for each layer, all layers had some predictive ability (Fig. 5E, $\rho > 0$). Deeper layers (5 to 10) tended to have better prediction than superficial layers (1 to 4). To establish a noise level for the V4 responses, we also predicted the norm and average distance for one session (day 1) with the V4 responses of another session (day 2), where the same images were shown each day. In other words, we used the V4 responses of day 2 as feature embeddings to predict V4 responses of day 1. The correlation of prediction was much higher

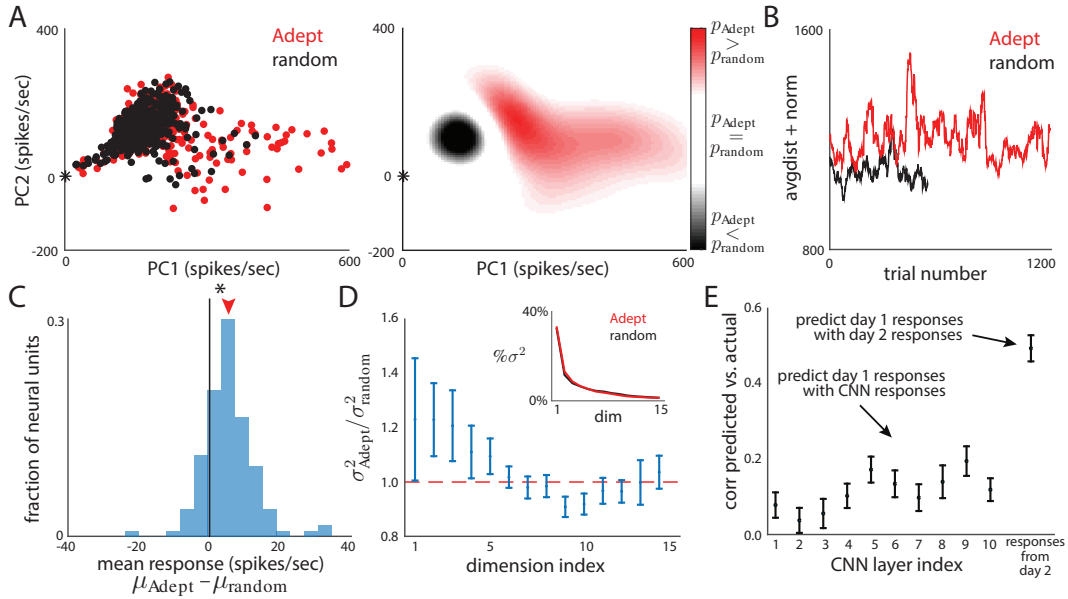

Figure 5: Closed-loop experiments in V4. *A*. Top 2 PCs of V4 responses to stimuli chosen by Adept and random selection (500 trials each). Left: scatter plot, where each dot represents the population response to one stimulus. Right: difference of kernel densities, $p_{\text{Adept}} - p_{\text{random}}$. Black * denotes a zero response for all neural units. *B*. Objective function evaluated across trials (one stimulus per trial) using V4 responses. Same data as in *A*. *C*. Difference in mean responses across neural units from 7 sessions. *D*. Ratio of eigenvalues for different PC dimensions. Error bars: $\pm$ s.e.m. *E*. Ability of different CNN layers to predict V4 responses. For comparison, we also used V4 responses from a different day to predict the same V4 responses. Error bars: $\pm$ s.d. across 100 runs.

($\rho \approx 0.5$) than that of any CNN layer ($\rho < 0.25$). This discrepancy indicates that finding feature embeddings that are more predictive of V4 responses is a way to improve Adept's performance.

## 5.3 Testing Adept for robustness to neural noise and overfitting

A potential concern for an adaptive method is that stimulus responses are susceptible to neural noise. Specifically, spike counts are subject to Poisson-like variability, which might not be entirely averaged away based on a finite number of stimulus repeats. Moreover, adaptation to stimuli and changes in attention or motivation may cause a gain factor to scale responses dynamically across a session [9]. To examine how Adept performs in the presence of noise, we first recorded a "ground-truth", spike-sorted dataset in which 2,000 natural images were presented (100 ms flashes, 5 to 30 repeats per image randomly presented throughout the session). We then re-ran Adept on simulated responses under three different noise models (whose parameters were fit to the ground truth data): a Poisson model ('Poisson noise'), a model that scales each response by a gain factor that varies independently from trial to trial [28] ('trial-to-trial gain'), and the same gain model but where the gain varies smoothly across trials ('slowly-drifting gain'). Because the drift in gain was randomly generated and may not match the actual drift in the recorded dataset, we also considered responses in which the drift was estimated across the recording session and added to the mean responses as their corresponding images were chosen ('recorded drift'). For reference, we also ran Adept on responses with no noise ('no noise'). To compare performance across the different settings, we computed the mean response and variance ratios between responses based on Adept and random selection (Fig. 6*A*). All settings showed better performance using Adept than random selection (Fig. 6*A*, all points above red-dashed line), and Adept performed best with no noise (Fig. 6, 'no noise' point at or above others). For a fair comparison, ratios were computed with the ground truth responses, where only the chosen images could differ across settings. These results indicate that, although Adept would benefit from removing neural noise, Adept continues to outperform random selection in the presence of noise.

Another concern for an adaptive method is overfitting. For example, when no relationship exists between the CNN feature embeddings and neural responses, Adept may overfit to a spurious stimulus-

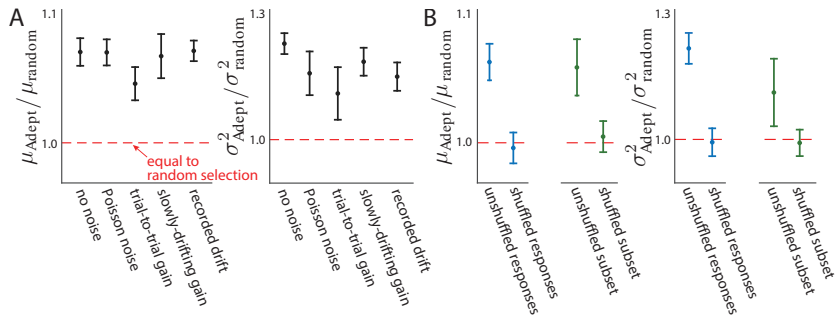

Figure 6: *A*. Adept is robust to neural noise. *B*. Adept shows no overfitting when responses are shuffled across images. Error bars: ± s.d. across 10 runs.

response mapping and perform worse than random selection. To address this concern, we performed two analyses using the same ground truth dataset as in Fig. 6*A*. For the first analysis, we ran Adept on the ground truth responses (choosing 500 of the 2,000 candidate images) to yield on average a 6% larger mean response and a 21% larger response scatter (average over top 5 PCs) than random selection (Fig. 6*B*, unshuffled responses). Next, to break any stimulus-response relationship, we shuffled all of the ground truth responses across images, and re-ran Adept. Adept performed no worse than random selection (Fig. 6*B*, shuffled responses, blue points on red-dashed line). For the second analysis, we asked if Adept focuses on the most predictable neurons to the detriment of other neurons. We shuffled all of the ground truth responses across images for half of the neurons, and ran Adept on the full population. Adept performed better than random selection for the subset of neurons with unshuffled responses (Fig. 6*B*, unshuffled subset), but no worse than random selection for the subset with shuffled responses (Fig. 6*B*, shuffled subset, green points on red-dashed line). Adept showed no overfitting in either scenario, likely because Adept cannot choose exceedingly similar images (i.e., differing by a few pixels) from its discrete candidate pool.

## 6 Discussion

Here we proposed Adept, an adaptive method for selecting stimuli to optimize neural population responses. To our knowledge, this is the first adaptive method to consider a population of neurons together. We found that Adept, using a population objective, is better able to optimize population responses than using a single-neuron objective to optimize the response of each neuron in the population (Fig. 4*C*). While Adept can flexibly incorporate different feature embeddings, we take advantage of the recent breakthroughs in deep learning and apply them to adaptive stimulus selection. Adept does not try to predict the response of each V4 neuron, but rather uses the similarity of CNN feature embeddings to different images to predict the similarity of the V4 population responses to those images.

Widely studied neural phenomena such as changes in responses due to attention [29] and trial-to-trial variability [30, 31] likely depend on mean response levels [32]. When recording from a single neuron, one can optimize to produce large mean responses in a straightforward manner. For example, one can optimize the orientation and spatial frequency of a sinusoidal grating to maximize a neuron's firing rate [9]. However, when recording from a population of neurons, identifying stimuli that optimize the firing rate of each neuron can be infeasible due to limited recording time. Moreover, neurons far from the sensory periphery tend to be more responsive to natural stimuli [33], and the search space for natural stimuli is vast. Adept is a principled way to efficiently search through a space of natural stimuli to optimize the responses of a population of neurons. Experimenters can run Adept for a recording session, and then present the Adept-chosen stimuli in subsequent sessions when probing neural phenomena.

A future challenge for adaptive stimulus selection is to generate natural images rather than selecting from a pre-existing pool of candidate images. For Adept, one could use a parametric model to generate natural images, such as a generative adversarial network [34], and optimize Eqn. 1 with gradient-based or Bayesian optimization.

## Acknowledgments

B.R.C. was supported by a BrainHub Richard K. Mellon Fellowship. R.C.W. was supported by NIH T32 GM008208, T90 DA022762, and the Richard K. Mellon Foundation. K.A. was supported by NSF GRFP 1747452. M.A.S. and B.M.Y. were supported by NSF-NCS BCS-1734901/1734916. M.A.S. was supported by NIH R01 EY022928 and NIH P30 EY008098. B.M.Y. was supported by NSF-NCS BCS-1533672, NIH R01 HD071686, NIH R01 NS105318, and Simons Foundation 364994.

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
