[Reviews · NeurIPS 2017]

Reviewer 1



The authors present an algorithm for selecting stimuli in a closed-loop experiment to maximize the variance in firing across a population of neurons (Adept). This is an important step in closed-loop stimulus selection, which have focused on learning the response properties of single neurons. The algorithm uses pre-computed features of the stimuli given by a neural network to predict firing rates to new stimuli. The methods were presented clearly and the stimulus selection is efficient enough to be useful to real experiments. There are a few limiting points of the Adept method (which may be addressed in future work). The applicability of this algorithm is limited, as mentioned by the authors, in that it depends on having a good set of features to describe the stimulus space. Such features may not always be available outside select areas such as V4. It would be valuable to explore the Adept algorithm's behavior when the features chosen do not provide a good basis for prediction spike rates (perhaps for only a portion of the population - would the algorithm learn less for those cells than random stimulus selection?). Additionally, the algorithm assumes a limited discrete set of stimuli. Finally, it is not clear how the stimulus selection may be affected by different types of noise (in particular, correlated noise across cells or time) in the responses.

Reviewer 2



An interesting method to speed up experiemntal design to reveal neural coding. The idea of using CNN to quantify stimulus similarity is interesting, but it also raises a paradox issue: if a CNN is able accurately predict the statistics of neuronal responses (such as in V4), then we do not have to do the experiment anymore, since analyzing the CNN data can tell us everything; on the other hand, if the CNN only partially or mistakenly measures the stimulus similarity that should be observed in the brain area recorded, then this mismatch will affect the experimental result. It would be interesting to see how this issue can be solved. But overall, this is a good attemp and is acceptable by NIPS.

Reviewer 3



This paper developed a simple and practical method for active learning (the Adept algorithm) a visual stimuli that maximizes a population cost function. The approach is elegant and intuitive. The technique was primarily discussed through the importance of choosing experimental conditions for experiments in which the experimenter must choose which kind of stimuli to present given a particular area of neural recording, and that Adept can adaptively find images to show next. Pretrained convolutional neural networks were used to estimate the response of unknown stimuli using kernel methods with a set of underlying features using Nadaraya-Watson regression. The first experiment used synthetic data from a separate CNN. The second, more interesting example performed a feedback experiment by showing stimuli to macaque’s while recording in V4, and was able to see an average 20% higher mean response from neurons with chosen stimuli. - The approach doesn't account for variability in the response. It would be interesting to see comparison under different noise models.